# EXPLOITING THE POTENTIAL OF SEQ2SEQ MODELS AS ROBUST FEW-SHOT LEARNERS

## ABSTRACT

In-context learning, which offers substantial advantages over fine-tuning, is predominantly observed in decoder-only models, while encoder-decoder (i.e., seq2seq) models excel in methods that rely on weight updates. Recently, a few studies have demonstrated the feasibility of few-shot learning with seq2seq models; however, this has been limited to tasks that align well with the seq2seq architecture, such as summarization and translation. Inspired by these initial studies, we provide a first-ever extensive experiment comparing the in-context few-shot learning capabilities of decoder-only and encoder-decoder models on a broad range of tasks. Furthermore, we propose two methods to more effectively elicit in-context learning ability in seq2seq models: objective-aligned prompting and a fusion-based approach. Remarkably, our approach outperforms a decoder-only model that is six times larger and exhibits significant performance improvements compared to conventional seq2seq models across a variety of settings. We posit that, with the right configuration and prompt design, seq2seq models can be highly effective few-shot learners for a wide spectrum of applications. [1]

## 1 INTRODUCTION

Recent studies have demonstrated that large language models can possess entirely different competencies, referred to as emergent abilities (Brown et al., 2020; Chowdhery et al., 2022; Rae et al., 2021; Wei et al., 2022b). The concept of emergent abilities in Large Language Models (LLMs), initially introduced by Wei et al. (2022b), posits that increasing the model size and dataset can lead to the sudden emergence of abilities such as in-context learning, complex reasoning, and common-sense reasoning. In-context learning, in particular, one of the distinct characteristics of LLMs, serves as a key metric for assessing their effectiveness.

In-context learning refers to the ability of the model to perform tasks by leveraging contextual information provided through prompts, without the need for additional training (i.e., without weight updates). Specifically, in the case of in-context few-shot learning, the model can generate suitable outputs for the desired target input by using a small number of examples. This offers a step beyond traditional frameworks, which typically require extensive training data and fine-tuning of the model. Instead, it presents a new paradigm where a single model can effortlessly perform new tasks without necessitating a separate training process.

The capability of in-context learning has been predominantly explored in decoder-only models, as the rapid evolution of pretrained models has mostly focused on these unidirectional architectures. However, the sequence-to-sequence (seq2seq) architecture, despite its significant advantage of encoding contexts without sacrificing bidirectionality, has not been extensively investigated regarding its potential for in-context learning capabilities (Sanh et al., 2022; Lester et al., 2021; Tay et al., 2022; Soltan et al., 2022; Patel et al., 2022; Wang et al., 2023).

Sanh et al. (2022) trained an encoder-decoder model using multitask prompts in a supervised manner, but this approach only enabled zero-shot generalization to new tasks. Soltan et al. (2022) primarily demonstrated the in-context abilities of seq2seq models for generative tasks such as summarization and translation, which are tasks inherently well-suited for seq2seq models. They also reported performance on natural language understanding (NLU) benchmarks but only in a zero-shot

---

[1]We will release the toolkit for the in-context evaluation of seq2seq models.

scenario, making it difficult to confirm their in-context learning proficiency across a wide range of tasks.

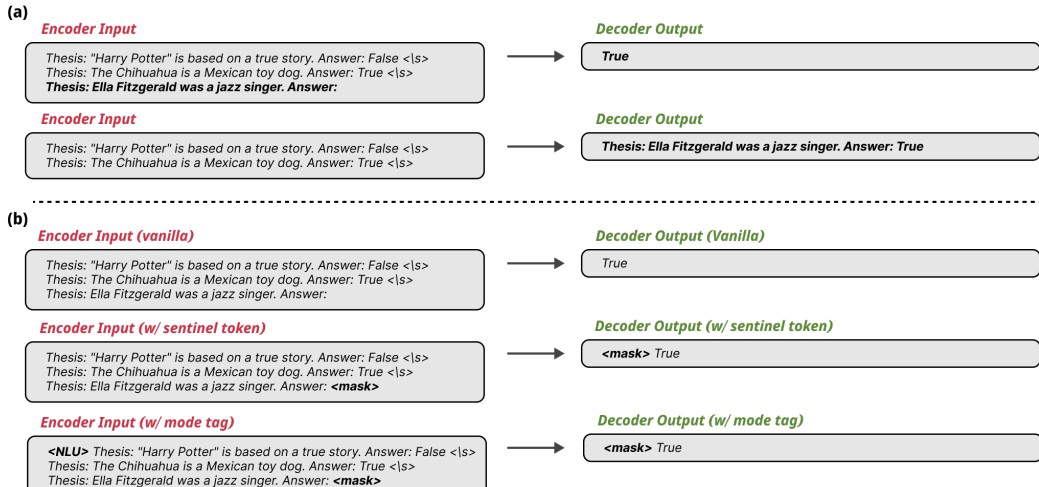

Figure 1: **Different prompting strategies for in-context learning.** **(a)** A target input can be placed to the encoder side, concatenated with few-shot examples, or decoder side standalone. **(b)** Examples of pretraining objective aligned prompt. In alignment with T5, the sentinel token is attached to the target output. In alignment with UL2, the mode tag is added as a prefix to the encoder input.

Motivated by the current landscape, for the first time, we thoroughly investigate the zero-shot to few-shot performance of seq2seq models across a wide range of evaluation sets. Our findings demonstrate that seq2seq models also serve as robust few-shot learners not only for generation tasks but also for understanding tasks, even outperforming their larger decoder-only counterparts (Zhang et al., 2022).

We conduct various experiments on how to structure the prompts in seq2seq models. While Raffel et al. (2020); Tay et al. (2022) popularized the usage of sentinel tokens during the pretraining stage of seq2seq models to optimize the denoising objective, the proper application of the sentinel tokens during the inference stage (i.e., in-context learning) is less well-established, and is often not even mentioned in most studies (Sanh et al., 2022; Tay et al., 2022). We find that aligning prompts with the pretraining objective yields up to a +20.5%p performance improvement on the SuperGLUE (Wang et al., 2019) benchmark.

Furthermore, we propose two fusion-based approaches that enhance the few-shot learning capability of encoder-decoder architecture models. The fundamental idea is to independently process each of the few-shot examples through an encoder and then merge these representations for decoding. This fusion-based approach offers several advantages. Firstly, it addresses the restricted maximum sequence length commonly found in seq2seq models (Raffel et al., 2020; Tay et al., 2022; Lester et al., 2021), which is shorter than decoder-only models, by leveraging encoding parallelization. Secondly, we achieve more efficient encoding of multiple examples by avoiding unnecessary bidirectional attention calculations between them. Our approach demonstrates significant performance improvements in seq2seq settings, surpassing the OPT 66B model (Zhang et al., 2022) across various tasks. Despite the absence of complete attention across the contexts of the few-shot prompts, our approach consistently outperforms traditional methods. Moreover, our methodologies effectively eliminate the permutation bias commonly observed in few-shot learning scenarios.

Overall, our work reveals the few-shot capability of seq2seq models, which has been undervalued compared to their zero-shot and fine-tuned counterparts. To summarize, our key contributions are: 1) we develop an in-context evaluation toolkit for seq2seq models and conduct extensive experiments to investigate the performance of seq2seq models in zero-shot to few-shot scenarios using fair criteria, 2) we exploit the potential of few-shot learning in encoder-decoder models by exploring prompting strategies and fusion-based approaches, and 3) we experimentally demonstrate that the seq2seq model can outperform the decoder-only model with 6 times larger parameters across diverse

| Model | *encoder* (1/5/10-shot) | *decoder* (1/5/10-shot) |
|---|---|---|
| T5 | **65.53/59.54/59.09** | 44.95/47.37/47.61 |
| T5-LM | **61.72/58.77/59.74** | 51.76/53.85/54.68 |
| T0 | **64.35/67.45/63.42** | 52.47/53.97/53.36 |
| UL2 | **60.05/58.85/60.02** | 51.46/57.41/59.14 |

Table 1: **Ablation on the placement of target input.** The *encoder* places the target input on the encoder side, whereas the *decoder* places it on the decoder side. Bold denotes the best score within each model and for the specific number of shots. The complete results are reported in Appendix A.

| Model | *vanilla* (0/1/5/10-shot) | *w/ sentinel* (0/1/5/10-shot) | *w/ mode tag* (0/1/5/10-shot) |
|---|---|---|---|
| T5 | 51.7/54.6/52.2/52.1 | **52.9/65.5/59.5/59.1** | - |
| T5-LM | 56.0/50.4/56.6/56.9 | **59.5/61.7/58.8/59.7** | - |
| T0 | **73.3/67.3**/60.4/56.8 | 72.2/64.4/**67.5/63.4** | - |
| UL2 | 52.4/50.1/48.4/49.3 | **58.8**/60.1/58.9/60.0 | 58.5/**62.2/61.2/62.3** |

Table 2: **Ablation on the usage of sentinel tokens and mode tags.** Figure 1-(b) depicts the structures of the *vanilla*, *w/ sentinel*, and *w/ mode tag* types. For *w/ mode* setting, the sentinel token is also utilized. Bold denotes the best score within each model and for the specific number of shots. Due to space constraints, scores are expressed up to the first decimal place. The complete results are reported in Appendix B.

datasets. In Section 7, we discuss the impact of these unprecedented findings on the evolution of LLMs.

## 2 EXPLORING PROMPTING STRATEGIES FOR SEQ2SEQ MODELS IN IN-CONTEXT LEARNING

We begin by demonstrating the different prompting strategies suitable for in-context learning of encoder-decoder models. Since decoder-only models share a unified left-to-right attention architecture (Radford & Narasimhan, 2018), we can naturally feed the demonstrations and target input in a sequential manner and perform generation to obtain the target output. For instance, in a 5-shot English-French translation task, the input sequence is formed by concatenating five English-to-French demonstrations, followed by target English input, with the model expected to generate the corresponding French translation as the target output.

However, encoder-decoder models process inputs and outputs independently from the encoder and decoder, respectively, which can result in varying compositions of demonstrations, target input, and target output, as shown in Figure 1-(a). The target input can be utilized either as an encoder input, resulting in the decoder generating only the target output, or as a decoder input, enabling the decoder to directly generate the answer conditioned on the target input. We evaluate this ablation on the SuperGLUE dataset, with 1, 5, and 10-shot settings using four representative encoder-decoder models. Table 1 shows that positioning the target input on the encoder side results in enhanced performance for all model types, with performance gains of up to +20.5%p. We interpret these results as follows: During the pretraining stage, the model generates output conditioned on bidirectionally encoded information from the encoder, instead of relying on the preceding textual information from the decoder. As a result, placing the target input on the encoder side shows better performance.

Additionally, recent state-of-the-art models include T5 (Raffel et al., 2020) and T5 variants (Lester et al., 2021; Sanh et al., 2022; Tay et al., 2022; Patel et al., 2022) suggest different training objectives for language modeling. Raffel et al. (2020); Lester et al. (2021); Sanh et al. (2022) employ denoising objectives to learn better bidirectional representations and utilize sentinel tokens to replace consecutive spans of tokens. Tay et al. (2022) proposed the model UL2, which additionally introduces the concept of mode switching with extra paradigm tags (i.e., [NLG],[NLU],[S2S]) that help the model learn suitable representations for a given task.

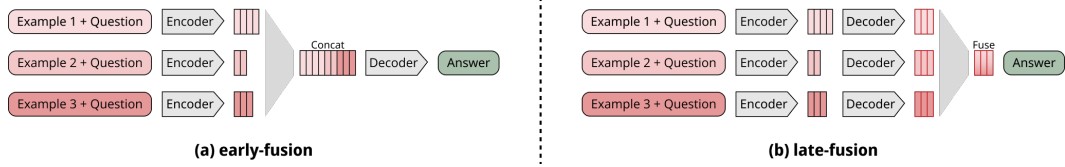

Figure 2: **An overview of the proposed approaches.** Each case is presenting a 3-shot setting.

We argue that, even when performing downstream tasks, it is crucial to design the prompt to resemble the pretraining scheme to achieve optimal performance. For instance, T5 manipulates inputs by masking consecutive tokens of the original text with a single sentinel token and then generates the target sequence by conditioning on that sentinel token. Thus, unlike decoder-only models that simply list examples as inputs, it is a more natural setting to prepend a sentinel token to the target for in-context learning using T5. As depicted in Figure 1-(b), we conduct experiments by incorporating a sentinel token and a mode tag into the vanilla prompt format. Since the mode tag is exclusively used in UL2, it is not applied to the remaining models. We assess the performance on the SuperGLUE datasets with various number of few-shot examples. In accordance with the pretraining methodologies of the models, we position the sentinel token at the end of the input and the model tag at the beginning. The results are presented in Table 2. Following the pretraining objective yields a maximum performance gain of up to +13%p. As expected, aligning with the pretraining objective produces the most favorable results, with a significant discrepancy in scores. It is noteworthy that even T5-LM (Lester et al., 2021) and T0 (Sanh et al., 2022), which have subsequent training phase without using sentinel token, demonstrate the positive impact of adding the sentinel token to the prompt. Throughout the following sections, we employ the optimal objective-aligned prompting strategies that we identified in this section, as the default configuration in all experiments.

## 3 FUSION-BASED APPROACHES FOR FEW-SHOT LEARNING

In this section, we address another factor that impairs the performance of few-shot learning and propose two fusion-based approaches to mitigate this problem. T5-family models (Raffel et al., 2020; Lester et al., 2021; Sanh et al., 2022; Tay et al., 2022) utilize relative position encoding (RPE) (Shaw et al., 2018), thereby enabling to process long input sequences as long as computing memory allows. However, as shown in Press et al. (2022), the extrapolation ability of T5-style RPE diminishes when the input length exceeds twice the maximum pretrained sequence length. That is, encoder-decoder models with shorter training sequence lengths of 512 (Raffel et al., 2020; Tay et al., 2022), compared to decoder-only models (Zhang et al., 2022; Brown et al., 2020; Black et al., 2022), often exhibit limited performance when dealing with long-context few-shot learning scenarios. Additionally, there is a challenge in terms of computational cost, which escalates quadratically as the number of shots increases. Another challenge to address is the presence of permutation bias. The order of demonstrations affects the positional embedding of tokens, resulting in different predictions even when the examples are identical but the order varies.

To address these problems, we propose a fusion-based approach, in which each demonstration is independently processed by the encoder. Afterwards, we merge them together at a specific stage. We hypothesize that the encoding of relations between demonstrations does not significantly impact in-context learning performance. We categorize this approach into two styles: *early-fusion* and *late-fusion*, differentiated by the stage at which the fusion occurs. These concepts are inspired by retrieval-based generation models. Lewis et al. (2020) propose Retrieval-Augmented Generation (RAG), where each retrieved document is integrated with the query and passed through the model independently, and the final output probability is calculated as a weighted sum of the probabilities of each output. Fusion-in-Decoder (Fid), introduced by Izacard & Grave (2021), differs from RAG in that the inputs passed through the encoder are concatenated before being sent to the decoder without any weighting. This allows the model to perform fusion at the decoder level. We apply these ideas to few-shot learning of seq2seq models by treating each few-shot example as a retrieved document. Therefore, we use the term *late-fusion* to refer to the RAG-style method that combines the decoder outputs, whereas the term *early-fusion* refers to the FiD-style method that combines the encoder outputs. The formulas are described below. The traditional seq2seq in-context learning, which we

refer to as the *original* method, constructs the encoder input by concatenating the target input $x$ with the few-shot examples $z$ to generate the target output $y$ in the decoder (Raffel et al., 2020; Tay et al., 2022). The probability of the target output for *original* method can be expressed as

$$P_{origin}(y|x, z) \approx \prod_{i}^{N} f_{\text{dec}}(y_i | f_{\text{enc}}(z_{1:k}, x), y_{1:i-1})$$

where $f_{enc}$ and $f_{dec}$ are encoder and decoder, respectively, and $k$ denotes the number of shots. Here, $y_i$ is the $i$-th target token and $y_{1:i-1}$ is the partial output. In the *late-fusion* approach, as depicted in Figure 2-(b), $j$-th example $z_j$ is concatenated to $x$ and each combination is utilized separately as the model input. At each token generation step $i$, the information from $k$ instances that have passed through the entire model is aggregated as follows:

$$P_{late}(y|x, z) \approx \prod_{i}^{N} \sum_{j}^{k} f_{\text{dec}}(y_i | f_{\text{enc}}(z_j, x), y_{1:i-1})$$

Unlike the original RAG model, in our few-shot settings, examples are obtained without going through the retrieval process. Therefore, we consider all examples to be equally retrieved, and no weighting is applied to individual examples.

Instead of combining the decoder outputs, *early-fusion* approach merges the information from the encoder output and passes it to the decoder, as shown in Figure 2-(a). All the last hidden states of the encoder, $h_1, h_2, ..., h_k$, where $h_j = f_{\text{enc}}(z_j, x)$, are concatenated sequentially and used for cross-attention,

$$P_{early}(y|x, z) \approx \prod_{i}^{N} f_{\text{dec}}(y_i | [h_1, h_2, ..., h_k], y_{1:i-1})$$

These fusion-based approaches successfully mitigate the extrapolation problem by maintaining a shorter length of the encoder input. Moreover, inference time can be reduced in proportion to the number of examples $k$ through batch processing. Further, our approaches can be interpreted as an ensemble strategy, where each example in a different encoder contributes a unique perspective, enhancing overall prediction accuracy. In the following sections, we experimentally demonstrate the capability of the seq2seq model as a robust few-shot learner by integrating the prompting strategies proposed in Section 2 with the fusion approaches presented in this section.

## 4 EXPERIMENTAL SETUP

We provide a detailed description of our systematic experimental setup in the following paragraphs. To ensure a fair evaluation, we employ identical prompt structures, evaluation tasks, scoring methods, prompt templates, and few-shot demonstrations across all baseline models.

### 4.1 BASELINE MODELS

To test the impacts of our proposed methods on seq2seq models, we employ T5 and its variants, including T5-LM, T0, and UL2, as baseline models. Among seq2seq baseline models, T0 is the only model fine-tuned with plenty of zero-shot multitask prompts, and UL2 contains 20B parameters while others contain 11B. For the comparison with decoder-only models, we adopt the following as decoder baselines: OPT (Zhang et al., 2022) and BLOOM (Workshop, 2023), with parameter sizes ranging from 7B to 66B. We provide detailed information about our baseline models in Appendix F.

### 4.2 EVALUATION TASKS AND SCORING METHODS

In Table 1 and 2, we evaluate the models on eight subtasks from SuperGLUE benchmarks to evaluate natural language understanding ability. For comprehensive evaluation across various types of tasks, we additionally assess the baseline models on 11 tasks selected by Sanh et al. (2022) including five subtasks from SuperGLUE, Hellaswag (Zellers et al., 2019), ANLI (Nie et al., 2020), Winogrande (Sakaguchi et al., 2019), and StoryCloze (Mostafazadeh et al., 2016). We report accuracy

| Model | Shot | RTE | CB | ANLI R1 | ANLI R2 | ANLI R3 | WSC | Winogrande | COPA | StoryCloze | HellaSwag* | WiC | average |
|---|---|---|---|---|---|---|---|---|---|---|---|---|---|
| BLOOM-7B | 5 | 59.35 | 51.43 | 33.66 | 33.92 | 34.22 | 36.54 | 65.02 | 75.60 | 70.83 | 46.32 | 50.63 | 50.68 |
| | 10 | 57.11 | 48.21 | 33.84 | 33.44 | 33.95 | 36.54 | 64.61 | 75.00 | 71.47 | 46.04 | 50.03 | 50.02 |
| | gpt3 | 57.47 | 57.14 | 33.40 | 33.64 | 34.62 | 36.54 | 64.75 | 78.80 | 71.96 | 46.62 | 50.03 | 51.36 |
| OPT-13B | 5 | 50.47 | 33.57 | 34.24 | 33.16 | 34.05 | 36.54 | 68.68 | 86.00 | 79.02 | 52.76 | 52.01 | 50.95 |
| | 10 | 49.17 | 42.14 | 34.04 | 33.52 | 35.00 | 36.54 | 68.29 | 86.40 | 79.70 | 52.54 | 52.79 | 51.83 |
| | gpt3 | 49.39 | 40.71 | 35.04 | 32.90 | 35.70 | 36.54 | 68.59 | 87.40 | 80.17 | 52.88 | 52.45 | 51.98 |
| OPT-30B | 5 | 63.61 | 38.21 | 30.75 | 33.64 | 32.53 | 38.65 | 69.49 | 84.80 | 78.87 | 54.82 | 52.63 | 52.55 |
| | 10 | 61.66 | 41.07 | 31.53 | 30.60 | 34.15 | 36.35 | 70.89 | 84.60 | 79.58 | 55.26 | 51.22 | 52.45 |
| | gpt3 | 62.67 | 58.57 | 31.78 | 31.56 | 32.72 | 36.54 | 70.75 | 86.60 | 80.57 | 55.70 | 52.23 | 54.52 |
| OPT-66B | 5 | 66.14 | 49.29 | 32.83 | 33.56 | 34.23 | 36.35 | 70.13 | 88.40 | 80.72 | 56.72 | 51.32 | 54.52 |
| | 10 | 63.68 | 56.79 | 32.50 | 33.88 | 34.45 | 36.54 | 70.31 | 88.40 | 81.53 | 56.62 | 51.54 | 55.11 |
| | gpt3 | 68.23 | 60.36 | 33.90 | 34.52 | 34.83 | 36.54 | 71.19 | 89.80 | 82.30 | 57.10 | 52.29 | 56.46 |
| T5-11B | 5 | 55.02 | 53.93 | 33.70 | 35.00 | 35.62 | 39.04 | 64.64 | 82.80 | 77.71 | 45.32 | 50.50 | 52.12 |
| | 10 | 56.10 | 51.43 | 33.66 | 33.88 | 33.80 | 37.88 | 64.66 | 82.00 | 77.67 | 44.64 | 52.10 | 51.62 |
| | gpt3 | 50.32 | 38.57 | 33.74 | 33.26 | 33.98 | 40.38 | 64.77 | 81.20 | 71.50 | 43.10 | 49.94 | 49.16 |
| T5-11B-*early* | 5 | 63.61 | 73.93 | 40.80 | 38.18 | 39.55 | 62.69 | 62.48 | 84.20 | 77.84 | 45.90 | 50.03 | **58.11** |
| | 10 | 63.39 | 77.86 | 41.18 | 38.32 | 39.58 | 65.19 | 62.23 | 84.80 | 77.56 | 45.80 | 50.00 | **58.72** |
| | gpt3 | 64.04 | 75.00 | 42.08 | 37.94 | 39.98 | 68.27 | 62.32 | 85.40 | 77.57 | 46.06 | 50.16 | **58.98** |
| T5-11B-*late* | 5 | 62.74 | 70.36 | 39.88 | 38.12 | 39.27 | 61.92 | 62.00 | 83.80 | 77.56 | 46.02 | 50.03 | 57.43 |
| | 10 | 63.32 | 73.93 | 40.16 | 38.32 | 39.57 | 63.85 | 62.13 | 83.80 | 77.31 | 45.78 | 50.00 | 58.01 |
| | gpt3 | 63.90 | 70.36 | 40.64 | 38.06 | 39.63 | 69.42 | 62.15 | 84.80 | 77.27 | 45.88 | 50.22 | 58.39 |

Table 3: **Comparison of our approach with various decoder models using minimal templates proposed by Gao et al. (2021).** Bold denotes the best average score, and underline denotes the best score within each task. T5 models with *early-* and *late-fusion* demonstrate superiority over decoder models in tasks such as CB and WSC, while exhibiting limitations in tasks such as Winogrande and Hellaswag. Tasks denoted with a star (*) indicate that 1K samples are evaluated due to their large size. The row labeled "gpt3" refers to the optimal number of shots for each task suggested by Brown et al. (2020). Detailed shot configurations are reported in Table 9. All the scores reported in this table are the average scores of 5 experiments with distinct random seeds.

as the metric, following the approach by Brown et al. (2020), and the model selects the option with the lowest cross-entropy loss among all the available multiple-choice options. Any form of normalization with respect to the target output is not applied. Please refer to Appendix H for justification regarding normalization, and to Appendix I for detailed explanations of each benchmark dataset.

We adopt two following generation tasks for the comprehensive evaluation: XSum (Narayan et al., 2018) and WebNLG (Moryossef et al., 2019), which are summarization and data-to-text tasks, respectively. All the samples are generated with greedy decoding, and the ROUGE (Lin, 2004) metric is used for the evaluation of generation tasks.

### 4.3 PROMPTS AND FEW-SHOT DEMONSTRATIONS

We employ prompt templates offered by Language Model Evaluation Harness (Gao et al., 2021), a widely adopted few-shot evaluation toolkit for decoder-only models. Additionally, for each generation task, we adopt a single template offered by PromptSource (Bach et al., 2022) since the templates for the generation tasks we utilize are not supported by Language Model Evaluation Harness. We conduct five experiments with different random seeds and present the averaged scores for Table 3 and Figure 3. For the other evaluations, we conduct a single experiment. For more detailed information about prompts and few-shot demonstrations, please refer to Appendix G.

## 5 EVALUATION RESULTS

### 5.1 SEQ2SEQ MODELS PERFORM BETTER THAN DECODER-ONLY MODELS

In Table 3, we evaluate both encoder-decoder and decoder-only models on a broad range of NLU benchmarks. We select T5 as the primary baseline model to apply our approaches because it is the most commonly used seq2seq pretrained model. Furthermore, T5-LM and T0 are models that have undergone further training with different objectives, making it unfair to compare them against other pretrained models. As shown in Table 3, T5 with both *early-* and *late-fusion* approaches achieve superior performance over OPT-66B model, which is six times larger in parameter size, with a

maximal margin of 3.6%p in average. Even with the best-shot configuration found by GPT-3 (Brown et al., 2020), our methods consistently exhibit superior performance compared to the decoder-only baselines in most cases, with a significant margin. Note that all the T5 variants reported in Table 3 already incorporate the objective-aligned prompt designs suggested in Section 2.

Complete results for the remaining seq2seq baselines (i.e., T5-LM, T0, and UL2) can be found in Appendix D. Additionally, we present the results of our method in comparison to other well-known decoder-only LLMs (i.e., GPT-3, PaLM, and GPT-NeoX) in Appendix C.

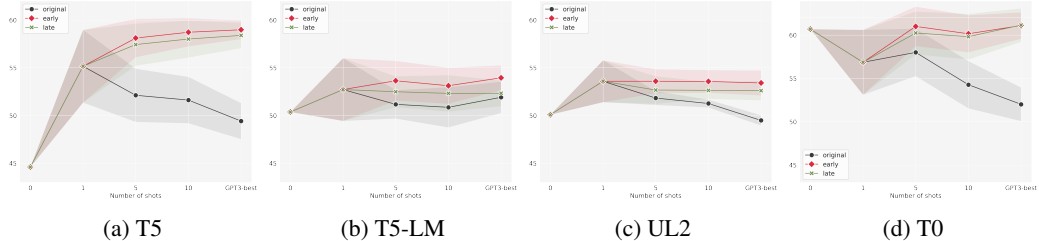

| (a) T5 | (b) T5-LM | (c) UL2 | (d) T0 |

Figure 3: **Comparison of in-context learning ability among seq2seq models by increasing the number of shots.** We experiment on the same tasks as presented in Table 3 and report the average accuracy and standard deviation results for four seq2seq baseline models.

## 5.2 ROBUSTNESS OF OUR APPROACH REGARDLESS OF THE BASELINE MODELS

In Figure 3, we examine the in-context learning abilities of different seq2seq baseline models and observe a trend that varies with the number of shots. Our approach demonstrates its effectiveness by consistently improving the performance when applied to all models. The black lines, which represent the *original* method, predominantly exhibit a decline in score in the few-shot setting, when compared to zero-shot or one-shot inference.

In contrast, the results obtained by applying our approach show a completely different tendency. The red and green lines, representing the application of *early-fusion* and *late-fusion*, respectively, exhibit an upward curve, indicating that the few-shot learning performance outperforms the 1-shot performance. The only exception is UL2 where the few-shot performance remains equal to the 1-shot performance.

Surprisingly, despite T0 being fine-tuned with zero-shot prompts, both of our approaches result in higher performance with the GPT-3 best-shot setting compared to the zero-shot result of the original T0 model. Considering that all four models were trained differently, this result reinforces the general applicability of our approach to seq2seq models, irrespective of how they were previously trained.

Additionally, we also observe that *early-fusion* consistently outperforms *late-fusion* by a small margin. Generally, the pretraining objective of encoder-decoder models is designed such that the decoder aggregates the outputs of the encoder. This process maximizes the joint probability of the input text sequence for the decoder, achieved through the encoder-decoder attention module. In this context, *early-fusion* implicitly selects examples that assist in resolving the test query by fully leveraging the encoder-decoder attention module. On the other hand, *late-fusion* does not differentiate whether individual examples aid in solving the test query; it simply aggregates all responses. In essence, *late-fusion* does not fully utilize the encoder-decoder attention module. Thus, we posit that the capability of the *early-fusion* to selectively prioritize certain examples, an ability absent in the *late-fusion*, is a major factor contributing to its superior performance. For detailed scores and standard deviations for each setting, please refer to Appendix D.

## 5.3 VALIDATION OF OUR APPROACH IN THE GENERATION TASKS

So far, we have validated our method across a range of natural language understanding tasks. In this subsection, we verify the robustness of our method by evaluating it on several generation tasks. As mentioned in Section 1, generation tasks, such as summarization and translation, align well with the objective of the encoder-decoder architecture and have been proven to be successful with seq2seq models. However, there is still a lack of organized evaluation of the few-shot learning ability on

| Model | 1-shot (R1/R2/RL) | 5-shot (R1/R2/RL) |
|---|---|---|
| T5* | 13.72/2.46/11.97 | 7.57/0.66/6.34 |
| T5 | 25.12/8.69/20.72 | 26.39/8.99/21.59 |
| T5-*early* | - | **30.31/11.55/25.10** |
| T5-*late* | - | 29.89/11.39/24.70 |

Table 4: **Evaluation on XSum dataset.** The asterisk(*) on the right side of the T5 denotes the case where the sentinel tokens are not used during inference time. R1, R2, and RL denotes ROUGE-1,2,L, respectively.

| Model | 1-shot (R1/R2/RL) | 32-shot (R1/R2/RL) |
|---|---|---|
| T5* | 18.63/7.98/16.28 | 0.25/0.13/0.24 |
| T5 | 39.13/23.44/32.61 | 41.40/22.63/34.09 |
| T5-*early* | - | **49.47/29.16/40.84** |
| T5-*late* | - | 48.47/28.54/40.60 |

Table 5: **Evaluation on WebNLG dataset.** The asterisk(*) on the right side of the T5 denotes the case where the sentinel tokens are not used during inference time. R1, R2, and RL denotes ROUGE-1,2,L, respectively.

| | OPT-13B | T5-*original* | T5-*early* | T5-*late* |
|---|---|---|---|---|
| average | 52.16 | 60.57 | **68.00** | 67.50 |
| std | 2.02 | 4.51 | **0.00** | **0.00** |

Table 6: **Evaluation of permutation bias for each method.** We experiment on an identical set of 5-shot demonstrations, with only a different order. Bold represents highest score for the average and lowest value for the standard deviation.

those tasks. We thus further evaluate our model on XSum and WebNLG datasets, in both one-shot and few-shot learning settings. Though studies from Tay et al. (2022) and Soltan et al. (2022) report the few-shot results of seq2seq models for XSum dataset, those results only encompass one-shot learning. Consequently, to the best of our knowledge, this is the first work that explores the results of "few"-shot learning for seq2seq models on the generation tasks.

The application of objective-aligned prompting and fusion-based approaches shows a remarkable improvement in the summarization task compared to the original T5 model, as shown in Table 4. A similar trend holds in WebNLG, a data-to-text task, as shown in Table 5. This implies that our proposed approach not only serves as a robust few-shot learner for understanding tasks but also for generation tasks. The analysis comparing decoder-only models in generative tasks can be found in Appendix E.

### 5.4 ANALYSIS OF THE PERMUTATION BIAS

One challenge of in-context few-shot learning is the potential variation in model predictions due to the order in which demonstrations are fed to the model. The *late-fusion* approach eliminates the so-called permutation bias problem, as it operates by fusing the probabilities of decoder outputs regardless of the order in which the inputs are fed. To verify the effectiveness of our methods in reducing permutation bias, we conduct experiments by reordering the few-shot examples. We select four tasks, CB, COPA, WSC, and WiC, from each of the four task taxonomies introduced by Sanh et al. (2022): natural language inference, sentence completion, coreference resolution, and word sense, to cover a diverse range of task types. We examine the permutation bias for a 5-shot setting and randomly sample 50 test sets from each task using the same random seed. In Table 6, we report the average and standard deviation for all 120 possible permutations with 5-shot examples for each task. Surprisingly, both the *early-fusion* and *late-fusion* methods demonstrate zero standard deviations for all tasks, thanks to the elimination of order relation on the encoder side. In the *early-fusion* method, there is a subtle variation in probability caused by the relative positional bias when the order of the examples is fused. However, this slight difference does not have any impact on the metric score. In contrast, the decoder-only baseline and the *original* encoder-decoder baseline show higher standard deviations with respect to the permutation of examples.

## 6 RELATED WORK

From UniLM (Dong et al., 2019) to BART (Lewis et al., 2019), T5, UL2, and AlexaTM (Soltan et al., 2022), encoder-decoder architecture models have continuously advanced and set new records by introducing novel denoising pretraining objectives. They excelled in tasks that require a com-

prehensive understanding of the input context and generate output based on it, such as translation, summarization, and other sequence-to-sequence tasks. However, this was particularly evident during the fine-tuning process. The seq2seq models have had limitations in handling sequences long enough, and it was only after the T5 series of papers that they were able to address this to some extent by incorporating relative position encoding. As a result, in-context few-shot learning, which necessitates a long encoder input length, has received relatively less attention. As a consequence, despite the recent explosive growth of LLMs, the size of encoder-decoder models has remained stagnant, with less than 20 billion parameters (Tay et al., 2022; Soltan et al., 2022). This stands in contrast to decoder-only models, which have reached the scale of hundreds of billions (Brown et al., 2020; Chowdhery et al., 2022; Zeng et al., 2022).

In-context learning has emerged as an alternative to the fine-tuning paradigm (Devlin et al., 2018; Radford & Narasimhan, 2018), which incurs expensive training costs, especially in the case of very large language models (Brown et al., 2020). According to Brown et al. (2020), pretrained decoder-only LLMs achieve proficiency in various tasks simply by prompting a few examples as input, without requiring any parameter updating. Unlike the many advancements of large decoder-only models (Wei et al., 2022b; Rae et al., 2021; Kim et al., 2021; Zhang et al., 2022; Touvron et al., 2023; Hao et al., 2022), the mainstream approach for encoder-decoder LLMs to adapt to a specific task remained supervised instruction-tuning (Sanh et al., 2022; Wei et al., 2022a; Longpre et al., 2023). Recently, a few studies attempted to explore in-context learning; UL2 and AlexaTM reported zero-shot results on the SuperGLUE dataset, and T0 utilized multitask prompted training to enhance zero-shot performance. Some of the studies employed techniques that emulate decoder-only models. Patel et al. (2022) utilized decoder-only-style sequential autoregressive prompting. Tay et al. (2022) mixed causal language modeling with denoising objectives. However, these approaches are restricted to particular setups and there has been a lack of structured results regarding few-shot learning.

## 7 DISCUSSION AND CONCLUSION

In this work, we demonstrate that the seq2seq model with proper adaptation enables few-shot learning across a broad range of tasks. To the best of our knowledge, no previous research has fully exploited the potential of seq2seq models as robust few-shot learners, outperforming decoder-only models, particularly on language understanding tasks. Our remarkable results are attributed to two factors: 1) the alignment of prompts with pretraining objectives and 2) the use of fusion-based methods that independently process examples, compensating for the objective incompatibility and structural shortcomings of seq2seq models, respectively. Through the carefully designed experiments, we verify that our approach exhibits a significant performance advantage under diverse conditions, including varying the number of shots, different baseline models, and a range of tasks, even with the complete removal of permutation bias. Another major contribution of our work is the provision of a unified in-context learning evaluation toolkit for seq2seq models. We have conducted controlled experiments to systematically compare and analyze the ability of existing seq2seq models — a process not previously undertaken. We plan to release the toolkit. Importantly, our findings rediscover the strengths of the encoder-decoder architectures and shed new light on their potential as conversational agents (e.g., GPT-4 (OpenAI, 2023)), which have been underestimated. We believe that the evolution of seq2seq models is still in progress, and our approach offers a substantial contribution toward maximizing their in-context learning capability.

## 8 LIMITATIONS

Apart from the reported results in this study, some discrepancies between benchmark performance and qualitative effectiveness still remain. The benchmark score itself cannot reflect linguistic abilities such as natural responses or reasoning, as recently showcased by GPT-4. Furthermore, even though we experiment on the baseline models with a similar size, there are still a few mismatches in terms of throughput, the composition of pretraining data, hyper-parameters, and the number of tokens used for pretraining. These differences are further discussed in Section F. In addition, there is a limitation in that our framework has not been applied to the BART-like pretrained encoder-decoder models. This is due to the fact that the large publicly available seq2seq models, which exceed a size of 10B, are only from the T5-family.

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

APPENDIX

We complement our paper with additional experimental results and miscellaneous details throughout this material. Section A provides the complete experimental results for Table 1 regarding the proper placement of the target input in the encoder-decoder model. Section B exhibits the impact of objective-aligned prompting for in-context learning, which complements Table 2. Moreover, we report the few-shot learning performance of other decoder-only LLMs that are not featured in the main paper, in Section C. Section D presents detailed numerical values utilized for reporting Figure 3. In Section E, we compare our methods with decoder-only models on generative tasks. Section F offers detailed information regarding the baseline models used in our experiments. Section G demonstrates examples of the prompts utilized in the generation tasks. Section H provides an explanation of how normalization is considered when measuring few-shot scores. Lastly, Section I gives explanations for each of the benchmark datasets.

## A    DETAILED RESULTS FOR TABLE 1

Table 1 verifies that the placement of a target input greatly impacts the in-context learning performance of seq2seq models. Table 7 provides detailed results of whether it is advantageous to have a target input in the encoder or the decoder, based on each task and each number of examples. In most cases, the target input given to the encoder produces significantly better results than that of the decoder. Although there are a few instances where the scores are improved when the input is given to the decoder for specific combinations of models and tasks (e.g., UL2 with RTE task, T5-LM with WSC task), these are small exceptions that make it difficult to identify trends.

## B    DETAILED RESULTS FOR TABLE 2

In Table 8, we present the complete results that complement Table 2. Without using sentinel tokens, T5 shows consistent scores across five out of eight tasks, regardless of the variation in the number of shots. However, this issue is resolved when sentinel tokens are used. Since T0 is further trained with the zero-shot prompts, we can observe that without the use of sentinel tokens, the score is initially higher at zero-shot results but significantly decreases as the number of demonstrations increases. On the other hand, when sentinel tokens are employed during inference, even though the highest scores are still observed in zero-shot scenarios, the performance is almost maintained as the number of demonstrations increases. Based on these observations, we conclude that aligning the prompt structure with the pretraining objective generally helps the seq2seq model better understand few-shot examples. Note that the scores reported in Table 8 are not the ones that applied our proposed methodologies like *early-* or *late-fusion*.

## C    COMPARING WITH OTHER DECODER LLMS

While numerous LLMs have their own benchmark evaluation scores reported, it is not guaranteed that these models are evaluated using the same baseline. Despite this potential discrepancy, to provide a point of reference, we present the results of our method alongside those of other well-known decoder-only LLMs in Table 10 including GPT-3, PaLM (Chowdhery et al., 2022) and GPT-NeoX (Black et al., 2022). All the records of decoder LLMs in Table 10 are from the paper. Compared to PaLM-8B, T5 with *early-fusion* achieves a higher average score. Although there is only a small overlap in tasks, T5-*early* beats GPT-NeoX on average and demonstrates much better performance in ANLI tasks compared to GPT-3. This indicates that our model performs well considering its size, as the throughput of the seq2seq model is comparable to that of a half-sized decoder-only model, as stated by Tay et al. (2022). In this regard, it is unfortunate that the absence of a larger encoder-decoder model hinders us from making an equivalent comparison with decoder models larger than 100B.

| Placement | Model | Shot | RTE | COPA | CB | WiC | WSC | BoolQ* | MultiRC* | RECORD* | average |
|---|---|---|---|---|---|---|---|---|---|---|---|
| encoder | T5 | 1 | 60.65 | 84.00 | 82.14 | 50.00 | 39.42 | 86.10 | 43.00 | 78.90 | **65.53** |
| | | 5 | 52.71 | 84.00 | 67.86 | 50.00 | 38.46 | 61.50 | 42.70 | 79.10 | **59.54** |
| | | 10 | 52.71 | 84.00 | 51.79 | 54.08 | 36.54 | 61.50 | 56.60 | 76.50 | **59.09** |
| | T5-LM | 1 | 64.02 | 83.00 | 52.57 | 50.00 | 36.54 | 74.30 | 46.20 | 85.50 | **61.72** |
| | | 5 | 47.29 | 78.00 | 57.14 | 50.00 | 44.23 | 61.50 | 46.90 | 85.10 | **58.77** |
| | | 10 | 47.29 | 79.00 | 53.57 | 50.00 | 60.58 | 61.50 | 43.60 | 82.40 | **59.74** |
| | T0 | 1 | 74.73 | 82.00 | 58.93 | 50.16 | 35.58 | 74.30 | 58.20 | 80.90 | **64.35** |
| | | 5 | 71.84 | 80.00 | 69.64 | 56.11 | 52.88 | 70.10 | 57.60 | 81.40 | **67.45** |
| | | 10 | 47.29 | 82.00 | 51.79 | 56.90 | 64.42 | 70.40 | 57.20 | 77.40 | **63.42** |
| | UL2 | 1 | 54.15 | 86.00 | 42.86 | 51.57 | 36.54 | 78.20 | 42.70 | 88.40 | **60.05** |
| | | 5 | 49.10 | 81.00 | 50.00 | 50.00 | 36.54 | 63.60 | 54.30 | 86.30 | **58.85** |
| | | 10 | 47.29 | 84.00 | 50.00 | 50.00 | 36.54 | 68.70 | 57.30 | 86.30 | **60.02** |
| decoder | T5 | 1 | 52.71 | 60.00 | 41.07 | 48.12 | 36.54 | 58.50 | 47.60 | 15.10 | 44.95 |
| | | 5 | 52.71 | 77.00 | 41.07 | 46.00 | 38.00 | 58.80 | 51.00 | 14.40 | 47.37 |
| | | 10 | 52.71 | 73.00 | 41.07 | 50.00 | 38.00 | 57.50 | 52.30 | 16.30 | 47.61 |
| | T5-LM | 1 | 51.99 | 74.00 | 46.43 | 50.00 | 36.54 | 63.00 | 42.60 | 49.50 | 51.76 |
| | | 5 | 47.29 | 72.00 | 48.21 | 47.00 | 62.00 | 61.50 | 42.70 | 50.10 | 53.85 |
| | | 10 | 47.29 | 75.00 | 51.79 | 46.00 | 62.00 | 61.50 | 42.70 | 51.20 | 54.68 |
| | T0 | 1 | 53.43 | 77.00 | 48.21 | 48.59 | 36.54 | 61.80 | 48.00 | 46.20 | 52.47 |
| | | 5 | 48.74 | 74.00 | 53.57 | 53.29 | 40.38 | 62.40 | 52.60 | 46.80 | 53.97 |
| | | 10 | 46.93 | 75.00 | 48.21 | 52.82 | 42.31 | 61.30 | 53.90 | 46.40 | 53.36 |
| | UL2 | 1 | 62.09 | 61.00 | 12.50 | 50.00 | 41.35 | 63.40 | 43.30 | 78.00 | 51.46 |
| | | 5 | 58.12 | 76.00 | 50.00 | 50.00 | 36.54 | 61.50 | 44.60 | 82.50 | 57.41 |
| | | 10 | 53.07 | 80.00 | 64.29 | 50.00 | 36.54 | 61.40 | 45.90 | 81.90 | 59.14 |

Table 7: **Detailed results for Table 1.** The asterisk(*) on the right side of the task indicates that due to the large size of the test dataset, evaluation is performed on a random sample of 1K instances from the test dataset. Bold denotes the best average score among different placements of the target input for each model and number of shots.

## D  ADDITIONAL RESULTS FOR TABLE 3 AND DETAILED RESULTS FOR FIGURE 3

In order to thoroughly analyze the performance of seq2seq baselines, we report the additive results evaluated on T5, T5-LM, T0, and UL2 for Table 3 in Table 11. Although we compare various decoder-only models with T5 (with our methodologies applied), other encoder-decoder models such as T5-LM and UL2 also achieve significant scores that approximate those of the OPT-30B model, which is much larger.

Table 11 is also the supplementary table for Figure 3, which depicts the average and standard deviation results for four seq2seq baselines for each methodology. The "std" values we report do not represent the standard deviation of scores across different tasks, but are calculated by averaging the standard deviation values across all tasks, with five seeds evaluated on each task. By applying our approaches, the average scores consistently increase for all models, while simultaneously reducing the standard deviation.

## E  COMPARISON WITH DECODER-ONLY MODELS ON GENERATIVE TASKS

The pretraining objectives of existing seq2seq models aren't ideally configured for free response generation; their focus was more on improving downstream task capabilities through span corruption and local reconstruction objectives, rather than on language modeling objectives. Consequently, for generative tasks, they might be less effective compared to decoder-only models. To illustrate this, we conducted comparisons with decoder-only baselines on representative generation tasks, XSum and WebNLG, with the results detailed in Table 12 and Table 13, respectively. We highlight the scores of both our model and the one exhibiting the best performance. Our best model, T5-early, outperforms OPT-13B in terms of ROUGE-2 and OPT-30B in terms of ROUGE-L on the XSum

dataset, despite having only 11B in size. However, it shows slightly lower overall performance compared to OPT-66B. For the WebNLG dataset, our model generally scores lower than decoder-only models. Notably, the significant difference in 5-shot performance between T5* and our proposed T5-early model indicates that our method meaningfully enhances the generation performance of the seq2seq model, irrespective of the pretrained language model's performance. With the improvements in the seq2seq model's pretraining objectives, we expect a considerable boost in performance.

## F  TRAINING DETAILS FOR BASELINE MODELS

Seq2seq models that we select as our baseline are trained in various manners. T5 is pretrained only with the denoising objective, utilizing a number of sentinel tokens that indicate where to be denoised. However, as the released version of T5 is a fine-tuned model, while T5.1.1 is trained solely on the C4 dataset with 1T tokens, we opt to utilize T5.1.1 as our T5 baseline. T5-LM is an additionally trained T5 model with a causal language modeling objective. T0 is a variant of T5-LM, which is fine-tuned with plenty of prompts for various tasks. It trained roughly 12B additional tokens compared to T5-LM. While Sanh et al. (2022) present three types of T5-architecture models (T0, T0+, and T0++) based on variations of fine-tuning datasets, we choose T0 to ensure that the datasets used in our experiments have not been previously encountered during the training process. Sentinel tokens were not used for further training both for T5-LM and T0. Finally, UL2 is a variant model of T5, pretrained with three types of denoising objectives: R-, S-, and X-denoising. All three objectives utilized the sentinel token, with "R" and "S" respectively corresponding to denoising and causal language modeling objectives. "X" represents the extreme version of both objectives. Similar to T5, it used the C4 dataset to train about 1 trillion tokens.

All four seq2seq baseline models are basically pretrained with sequence lengths of 512 (Raffel et al., 2020; Tay et al., 2022) or 1,024 (Sanh et al., 2022), which account for a quarter or half of the typical decoder LLMs (Brown et al., 2020; Zhang et al., 2022; Rae et al., 2021). Therefore, the seq2seq model faces a disadvantage when the number of examples used for few-shot evaluation increases, as it lacks exposure to long-length inputs during training.

According to Zhang et al. (2022), the OPT models are trained on approximately up to 300B tokens, including the Pile (Gao et al., 2020) and the datasets used for RoBERTa (Liu et al., 2019) pretraining. In our experiments, we evaluate three different sizes of OPT models (13B, 30B, and 66B), which are up to 6 times larger than the T5 baseline model. Additionally, to compare with decoder models smaller than 13B, we also evaluate the BLOOM-7B model, which is pretrained on approximately 340 billion tokens from the ROOTS corpus.

## G  DETAILED INFORMATION ABOUT PROMPTS AND FEW-SHOT DEMONSTRATIONS

For the natural language generation tasks in Table 4 and Table 5, we adopt one of the prompts from the PromptSource (Bach et al., 2022) since Language Model Evaluation Harness does not provide templates for XSum and WebNLG. We manually select one of the prompts from the PromptSource that best describes the task. We provide one-shot prompting samples utilizing the selected template in Figure 4.

With respect to the few-shot demonstrations, we distinguish between two settings where the demonstrations used for each prediction are maintained the same, referred to as the "fixed" setting, or randomly sampled at each time, referred to as the "non-fixed" setting. For the generation tasks, we conduct experiments with a "non-fixed" setting, otherwise utilize a "fixed" setting.

## H  NORMALIZATION TO THE SCORES

As mentioned in Section 4.2, we do not apply any normalization to the output losses, with respect to the token or string length of the output sequences. In the case of HellaSwag, applying normalization often leads to a higher score. However, it may not hold true for other tasks. To ensure consistent evaluation across all datasets, we decide to adopt a unified approach without normalization.

| Prompt | Model | Shot | RTE | CB | WSC | COPA | WiC | Boolq* | Multirc* | ReCoRD* | average |
|---|---|---|---|---|---|---|---|---|---|---|---|
| vanilla | T5 | 0 | 52.71 | 41.07 | 36.54 | 59.00 | 50.00 | 61.50 | 42.70 | 70.10 | 51.70 |
| | | 1 | 52.71 | 39.29 | 36.54 | 79.00 | 50.00 | 61.50 | 42.70 | 75.10 | 54.60 |
| | | 5 | 52.71 | 41.07 | 36.54 | 61.00 | 50.00 | 61.50 | 42.70 | 71.90 | 52.18 |
| | | 10 | 52.71 | 41.07 | 36.54 | 63.00 | 50.00 | 61.50 | 42.70 | 69.20 | 52.09 |
| | T5-LM | 0 | 52.71 | 26.79 | 63.46 | 71.00 | 50.31 | 40.80 | 57.30 | 85.60 | 56.00 |
| | | 1 | 47.29 | 17.86 | 36.54 | 63.00 | 44.83 | 61.60 | 49.60 | 82.20 | 50.36 |
| | | 5 | 46.57 | 35.71 | 63.46 | 60.00 | 50.00 | 61.50 | 57.40 | 78.20 | 56.61 |
| | | 10 | 47.29 | 41.07 | 63.46 | 60.00 | 50.00 | 61.50 | 57.30 | 74.50 | 56.89 |
| | T0 | 0 | 83.75 | 76.79 | 73.08 | 76.00 | 50.47 | 73.00 | 72.70 | 80.60 | **73.30** |
| | | 1 | 74.73 | 55.36 | 68.27 | 78.00 | 50.00 | 65.70 | 66.60 | 80.10 | **67.34** |
| | | 5 | 47.29 | 53.57 | 63.46 | 75.00 | 50.00 | 56.80 | 57.30 | 79.50 | 60.37 |
| | | 10 | 47.29 | 50.00 | 63.46 | 70.00 | 50.00 | 38.50 | 57.30 | 77.60 | 56.77 |
| | UL2 | 0 | 52.71 | 21.43 | 35.58 | 71.00 | 48.75 | 60.60 | 43.20 | 85.90 | 52.39 |
| | | 1 | 52.71 | 8.93 | 37.50 | 64.00 | 49.84 | 60.10 | 42.70 | 85.00 | 50.10 |
| | | 5 | 52.71 | 8.93 | 38.46 | 59.00 | 50.31 | 50.50 | 43.10 | 84.00 | 48.38 |
| | | 10 | 52.71 | 8.93 | 36.54 | 59.00 | 49.84 | 61.40 | 42.70 | 83.30 | 49.30 |
| w/ sentinel | T5 | 0 | 54.87 | 33.93 | 41.35 | 58.00 | 44.67 | 71.60 | 44.90 | 73.80 | **52.89** |
| | | 1 | 60.65 | 82.14 | 39.42 | 84.00 | 50.00 | 86.10 | 43.00 | 78.90 | **65.53** |
| | | 5 | 52.71 | 67.86 | 38.46 | 84.00 | 50.00 | 61.50 | 42.70 | 79.10 | **59.54** |
| | | 10 | 52.71 | 51.79 | 36.54 | 83.00 | 54.08 | 61.50 | 56.60 | 76.50 | **59.09** |
| | T5-LM | 0 | 62.45 | 48.21 | 36.54 | 71.00 | 50.00 | 75.20 | 46.10 | 86.40 | **59.49** |
| | | 1 | 64.62 | 53.57 | 36.54 | 83.00 | 50.00 | 74.30 | 46.20 | 85.50 | **61.72** |
| | | 5 | 47.29 | 57.14 | 44.23 | 78.00 | 50.00 | 61.50 | 46.90 | 85.10 | **58.77** |
| | | 10 | 47.29 | 53.57 | 60.58 | 79.00 | 50.00 | 61.50 | 43.60 | 82.40 | **59.74** |
| | T0 | 0 | 81.95 | 78.57 | 53.85 | 82.00 | 55.64 | 79.40 | 65.20 | 80.80 | 72.18 |
| | | 1 | 74.73 | 58.93 | 35.58 | 82.00 | 50.16 | 74.30 | 58.20 | 80.90 | 64.35 |
| | | 5 | 71.84 | 69.64 | 52.88 | 80.00 | 56.11 | 70.10 | 57.60 | 81.50 | **67.46** |
| | | 10 | 47.29 | 51.79 | 64.42 | 82.00 | 56.90 | 70.40 | 57.20 | 77.40 | **63.42** |
| | UL2 | 0 | 51.62 | 33.93 | 53.85 | 69.00 | 51.41 | 76.60 | 49.70 | 84.60 | **58.84** |
| | | 1 | 54.15 | 42.86 | 36.54 | 86.00 | 51.57 | 78.20 | 42.70 | 88.40 | 60.05 |
| | | 5 | 49.10 | 50.00 | 36.54 | 81.00 | 50.00 | 63.60 | 54.30 | 86.30 | 58.85 |
| | | 10 | 47.29 | 50.00 | 36.54 | 84.00 | 50.00 | 68.70 | 57.30 | 86.30 | 60.02 |
| w/ mode | UL2 | 0 | 53.79 | 8.93 | 47.12 | 87.00 | 53.13 | 80.30 | 50.40 | 87.20 | 58.48 |
| | | 1 | 48.01 | 44.64 | 46.15 | 86.00 | 54.70 | 88.40 | 42.90 | 86.90 | **62.21** |
| | | 5 | 48.38 | 50.00 | 36.54 | 83.00 | 50.47 | 86.30 | 48.40 | 86.30 | **61.17** |
| | | 10 | 47.29 | 50.00 | 36.54 | 86.00 | 50.00 | 86.30 | 57.30 | 84.90 | **62.29** |

Table 8: **Detailed results for Table 2.** The asterisk(*) on the right side of the task indicates that due to the large size of the test dataset, evaluation is performed on a random sample of 1K instances from the test dataset. For *w/ mode* setting, the sentinel token is also utilized. Bold denotes the best average score among *vanilla*, *w/ sentinel*, and *w/ mode* approaches, for the same model and the same number of shots.

| Task | RTE | CB | ANLI1 | ANLI2 | ANLI3 | WSC | Winogrande | COPA | StoryCloze | HellaSwag | WiC |
|---|---|---|---|---|---|---|---|---|---|---|---|
| shots | 32 | 32 | 50 | 50 | 50 | 32 | 16 | 32 | 70 | 20 | 32 |

Table 9: **The number of shots with the highest score reported in GPT-3.** For each task, the length of concatenated examples approximates the sequence length of the pretrained GPT-3 model. Consequently, conducting few-shot learning with such settings is disadvantageous for seq2seq models, which are pretrained on shorter sequence lengths.

Additionally, in Brown et al. (2020), a pre-normalization step is conducted where calibration is applied to each calculation of scores. With the same context mentioned in the paragraph above, we do not implement any preceding calibration to calculate the output scores.

| Model | RTE | CB | ALNI1 | ANLI2 | ANLI3 | WSC | Winogrande | COPA | StoryCloze | HellaSwag | WiC | **average** |
|---|---|---|---|---|---|---|---|---|---|---|---|---|
| PaLM-8B | 56.7 | 57.1 | 29.8 | 32.5 | 32.7 | 83.2 | 70.1 | 86.0 | 81.5 | 68.6 | 52.4 | 59.1 |
| PaLM-540B | 81.2 | 89.3 | 56.9 | 56.1 | 51.2 | 89.5 | 85.1 | 95.0 | 89.0 | 83.8 | 64.6 | **76.5** |
| GPT-NeoX 20B | - | - | 32.2 | 33.1 | 34.6 | 38.5 | 68.3 | - | - | 53.8 | - | (43.4) |
| GPT3 175B | 72.9 | 82.1 | 36.8 | 34.0 | 40.2 | 75.0 | 77.7 | 92.0 | 87.7 | 79.3 | 55.3 | 66.6 |
| T5-*early* 11B | 64.0 | 77.9 | 42.1 | 38.3 | 40.0 | 68.3 | 62.5 | 83.8 | 75.3 | 50.4 | 50.2 | 59.3(50.3) |

Table 10: **Comparison with other decoder LLMs for various NLU tasks.** The scores for PaLM and GPT-NeoX are 5-shot results. And the scores for GPT3 and T5-*early* are the highest scores among the reported ones for each task. The average scores inside the parentheses indicate the average scores of tasks where the GPT-NeoX report. Note that normalization is applied to the scores for PaLM and GPT3, which might be advantageous compared to when it is not applied.

| Model | Method | Shot | RTE | CB | ANLI1 | ANLI2 | ANLI3 | WSC | Winogrande | COPA | StoryCloze | HellaSwag* | WiC | average | std |
|---|---|---|---|---|---|---|---|---|---|---|---|---|---|---|---|
| T5-LM | *original* | 0 | 62.45 | 48.21 | 37.70 | 34.90 | 37.25 | 36.54 | 55.09 | 71.00 | 72.85 | 48.02 | 50.00 | 50.37 | 0.05 |
| | | 1 | 56.68 | 57.86 | 34.96 | 34.40 | 35.87 | 41.92 | 61.15 | 81.20 | 74.83 | 50.70 | 50.38 | 52.72 | 3.28 |
| | | 5 | 47.65 | 56.07 | 32.66 | 34.06 | 33.97 | 43.65 | 61.29 | 77.80 | 75.34 | 50.38 | 49.97 | 51.17 | 1.48 |
| | | 10 | 47.22 | 47.50 | 32.12 | 32.96 | 33.07 | 52.31 | 61.42 | 79.20 | 74.47 | 49.18 | 50.00 | 50.86 | 2.10 |
| | | gpt3 | 47.08 | 58.93 | 32.86 | 32.42 | 34.47 | 62.12 | 61.04 | 77.60 | 67.26 | 47.28 | 50.00 | 51.91 | 1.63 |
| | *early-fusion* | 5 | 53.86 | 56.07 | 35.62 | 34.84 | 36.95 | 52.50 | 61.93 | 82.20 | 75.53 | 50.52 | 49.97 | 53.64 | 2.07 |
| | | 10 | 53.29 | 53.21 | 35.82 | 34.58 | 36.47 | 50.96 | 61.86 | 81.80 | 75.60 | 50.60 | 50.13 | 53.12 | 1.88 |
| | | gpt3 | 52.20 | 53.21 | 35.82 | 34.46 | 36.90 | 60.19 | 61.82 | 82.80 | 75.25 | 50.44 | 50.31 | **53.95** | 1.30 |
| | *late-fusion* | 5 | 54.22 | 56.79 | 35.60 | 34.48 | 36.93 | 40.38 | 61.18 | 81.60 | 75.51 | 50.30 | 50.34 | 52.49 | 1.61 |
| | | 10 | 53.14 | 53.93 | 35.74 | 34.26 | 36.78 | 43.46 | 61.17 | 81.20 | 75.29 | 50.28 | 50.22 | 52.32 | 1.91 |
| | | gpt3 | 53.29 | 54.29 | 35.48 | 33.96 | 36.67 | 42.88 | 61.37 | 81.60 | 75.02 | 50.20 | 50.91 | 52.33 | 1.34 |
| T5 | *original* | 0 | 54.87 | 33.93 | 35.50 | 33.10 | 33.42 | 41.35 | 52.33 | 58.00 | 65.53 | 38.02 | 44.67 | 44.61 | 0.08 |
| | | 1 | 62.38 | 67.50 | 37.52 | 38.20 | 40.10 | 43.46 | 61.66 | 83.60 | 77.00 | 45.68 | 49.62 | 55.16 | 3.80 |
| | | 5 | 55.02 | 53.93 | 33.70 | 35.00 | 35.62 | 39.04 | 64.64 | 82.80 | 77.71 | 45.32 | 50.50 | 52.12 | 2.78 |
| | | 10 | 56.10 | 51.43 | 33.66 | 33.88 | 33.80 | 37.88 | 64.66 | 82.00 | 77.67 | 44.64 | 52.10 | 51.62 | 2.42 |
| | | gpt3 | 50.32 | 38.57 | 33.74 | 33.26 | 33.98 | 40.38 | 64.77 | 81.20 | 71.50 | 43.10 | 49.94 | 49.16 | 1.99 |
| | *early-fusion* | 5 | 63.61 | 73.93 | 40.80 | 38.18 | 39.55 | 62.69 | 62.48 | 84.20 | 77.84 | 45.90 | 50.03 | 58.11 | 2.00 |
| | | 10 | 63.39 | 77.86 | 41.18 | 38.32 | 39.58 | 65.19 | 62.23 | 84.80 | 77.56 | 45.80 | 50.00 | 58.72 | 1.47 |
| | | gpt3 | 64.04 | 75.00 | 42.08 | 37.94 | 39.98 | 68.27 | 62.32 | 85.40 | 77.57 | 46.06 | 50.16 | **58.98** | 0.97 |
| | *late-fusion* | 5 | 62.74 | 70.36 | 39.88 | 38.12 | 39.27 | 61.92 | 62.00 | 83.80 | 77.56 | 46.02 | 50.03 | 57.43 | 2.20 |
| | | 10 | 63.32 | 73.93 | 40.16 | 38.32 | 39.57 | 63.85 | 62.13 | 83.80 | 77.31 | 45.78 | 50.00 | 58.01 | 1.93 |
| | | gpt3 | 63.90 | 70.36 | 40.64 | 38.06 | 39.63 | 69.42 | 62.15 | 84.80 | 77.27 | 45.88 | 50.22 | 58.39 | 1.32 |
| T0 | *original* | 0 | 81.95 | 78.57 | 45.20 | 40.40 | 42.42 | 53.85 | 60.93 | 82.00 | 75.95 | 50.56 | 55.64 | 60.68 | 0.13 |
| | | 1 | 74.08 | 72.50 | 41.52 | 37.20 | 38.73 | 41.73 | 61.10 | 81.20 | 75.09 | 50.56 | 51.76 | 56.86 | 3.74 |
| | | 5 | 67.94 | 76.07 | 39.42 | 36.28 | 38.53 | 60.19 | 61.56 | 81.60 | 73.18 | 49.06 | 54.20 | 58.00 | 2.73 |
| | | 10 | 53.57 | 60.36 | 36.06 | 34.80 | 35.12 | 62.88 | 60.36 | 81.60 | 71.28 | 47.92 | 52.98 | 54.27 | 2.72 |
| | | gpt3 | 47.29 | 63.21 | 33.30 | 33.28 | 33.15 | 63.85 | 60.17 | 79.20 | 61.98 | 46.10 | 50.63 | 52.01 | 1.93 |
| | *early-fusion* | 5 | 81.16 | 76.43 | 44.36 | 37.86 | 42.60 | 63.65 | 61.86 | 82.20 | 75.25 | 50.40 | 55.27 | 61.00 | 2.28 |
| | | 10 | 79.78 | 68.21 | 44.26 | 38.58 | 41.57 | 64.23 | 61.86 | 82.20 | 75.05 | 50.30 | 55.64 | 60.15 | 2.13 |
| | | gpt3 | 81.95 | 70.71 | 44.38 | 37.90 | 41.90 | 68.08 | 62.18 | 83.80 | 74.86 | 50.10 | 56.21 | 61.10 | 1.52 |
| | *late-fusion* | 5 | 81.08 | 77.50 | 44.18 | 37.76 | 42.07 | 55.00 | 62.12 | 82.60 | 75.34 | 50.62 | 54.33 | 60.24 | 2.47 |
| | | 10 | 80.72 | 70.71 | 44.54 | 38.34 | 42.42 | 54.81 | 62.16 | 82.00 | 75.09 | 50.80 | 56.39 | 59.82 | 2.62 |
| | | gpt3 | 81.37 | 74.29 | 44.52 | 38.12 | 41.98 | 64.42 | 62.23 | 84.40 | 74.89 | 50.38 | 56.02 | **61.15** | 1.92 |
| UL2 | *original* | 0 | 53.79 | 8.93 | 33.10 | 33.00 | 34.92 | 47.12 | 64.17 | 87.00 | 78.41 | 57.46 | 53.13 | 50.09 | 0.08 |
| | | 1 | 54.87 | 49.29 | 34.36 | 33.76 | 33.32 | 47.50 | 63.96 | 85.00 | 78.31 | 56.86 | 52.29 | **53.59** | 2.19 |
| | | 5 | 49.03 | 50.00 | 33.32 | 33.52 | 33.50 | 36.54 | 66.35 | 84.20 | 77.66 | 55.96 | 50.09 | 51.83 | 0.72 |
| | | 10 | 47.51 | 50.00 | 33.30 | 33.30 | 32.97 | 36.54 | 64.67 | 84.20 | 76.77 | 54.60 | 50.00 | 51.26 | 0.41 |
| | | gpt3 | 47.29 | 50.00 | 33.50 | 33.30 | 33.12 | 36.54 | 63.84 | 82.00 | 69.06 | 51.64 | 50.13 | 49.49 | 0.53 |
| | *early-fusion* | 5 | 55.81 | 47.50 | 33.88 | 32.48 | 32.60 | 47.88 | 65.35 | 84.80 | 79.20 | 57.12 | 52.92 | **53.59** | 1.25 |
| | | 10 | 54.95 | 47.50 | 34.12 | 31.72 | 32.22 | 49.23 | 65.02 | 85.60 | 79.17 | 56.98 | 52.79 | 53.57 | 1.18 |
| | | gpt3 | 54.37 | 47.14 | 33.94 | 31.52 | 31.58 | 49.42 | 65.13 | 86.00 | 79.14 | 56.92 | 52.35 | 53.41 | 1.30 |
| | *late-fusion* | 5 | 52.78 | 46.07 | 33.24 | 32.62 | 32.90 | 44.81 | 65.89 | 84.20 | 78.97 | 56.80 | 51.03 | 52.67 | 1.44 |
| | | 10 | 53.36 | 45.71 | 33.46 | 31.94 | 32.62 | 45.58 | 65.59 | 84.40 | 78.80 | 56.84 | 50.75 | 52.64 | 0.89 |
| | | gpt3 | 53.07 | 46.07 | 33.30 | 31.58 | 32.85 | 46.54 | 65.73 | 83.60 | 78.92 | 56.90 | 50.19 | 52.61 | 1.01 |

Table 11: **Detailed results for Figure 3.** The asterisk(*) on the right side of the task indicates that due to the large size of the test dataset, evaluation is performed on a random sample of 1K instances from the test dataset. The underlined values represent the highest scores within each model for each task, while the bold values represent the settings where the average scores for all the tasks are highest within each model.

## I DETAILED EXPLANATIONS ABOUT EACH BENCHMARK DATASET

As outlined in Section 4.2, we adopted the SuperGlue benchmark along with four additional datasets to evaluate natural language understanding ability, and XSum and WebNLG for the generation task. The following provides a detailed description of each dataset.

- **BoolQ (Boolean Questions):** BoolQ is a dataset comprising natural yes/no questions derived from Google searches, with each question paired with a relevant Wikipedia passage.
- **CB (CommitmentBank):** CB is a textual entailment task in which models must judge if a given sentence entails, contradicts, or is neutral to a hypothesis. This task tests a model's ability to understand implications and contradictions in texts.

| Model | 1-shot (R1/R2/RL) | 5-shot (R1/R2/RL) |
|---|---|---|
| T5* | 13.72/2.46/11.97 | 7.57/0.66/6.34 |
| T5 | 25.12/8.69/20.72 | 26.39/8.99/21.59 |
| T5-*early* | - | **30.31/11.55/25.10** |
| BLOOM-7B | 21.50/4.75/16.33 | 21.96/5.06/17.04 |
| OPT-13B | 28.37/9.93/22.46 | 31.32/11.52/24.72 |
| OPT-30B | 27.61/10.06/22.20 | 31.54/12.06/25.19 |
| OPT-66B | 29.31/10.64/23.45 | **32.52/12.86/26.19** |

Table 12: **Evaluation on XSum dataset.** The asterisk(*) on the right side of the T5 denotes the case where the sentinel tokens are not used during inference time. R1, R2, and RL denotes ROUGE-1,2,L, respectively.

| Model | 1-shot (R1/R2/RL) | 32-shot (R1/R2/RL) |
|---|---|---|
| T5* | 18.63/7.98/16.28 | 0.25/0.13/0.24 |
| T5 | 39.13/23.44/32.61 | 41.40/22.63/34.09 |
| T5-*early* | - | **49.47/29.16/40.84** |
| BLOOM-7B | 62.77/38.54/51.19 | 68.07/43.97/56.1 |
| OPT-13B | 55.03/33.04/45.53 | 66.26/43.44/55.85 |
| OPT-30B | 58.00/35.35/47.78 | 66.57/43.07/55.22 |
| OPT-66B | 61.44/37.54/50.53 | **68.77/45.37/57.75** |

Table 13: **Evaluation on WebNLG dataset.** The asterisk(*) on the right side of the T5 denotes the case where the sentinel tokens are not used during inference time. R1, R2, and RL denotes ROUGE-1,2,L, respectively.

- COPA (Choice of Plausible Alternatives): COPA is a causality detection task where models are required to determine either the cause or effect from two possible options for a given statement.

- MultiRC (Multi-Sentence Reading Comprehension): MultiRC involves answering questions about a paragraph, with each question having multiple correct answers. It assesses the model's ability to understand and integrate information from multiple sentences.

- ReCoRD (Reading Comprehension with Commonsense Reasoning Dataset): ReCoRD is a multiple-choice QA task that challenges models to predict a masked-out entity from a list of possible entities in the provided passage.

- RTE (Recognizing Textual Entailment): RTE requires models to determine whether one sentence logically follows from another.

- WiC (Word-in-Context): WiC is a binary classification task for word sense disambiguation, requiring models to determine if a polysemous word is used in the same sense across two different sentences.

- WSC (Winograd Schema Challenge): WSC is a coreference resolution task that involves selecting the correct referent of the pronoun in a sentence, requiring commonsense reasoning and everyday knowledge.

- HellaSwag: HellaSwag is a benchmark dataset for testing commonsense NLI that is particularly challenging for state-of-the-art models, yet its questions are trivial for humans.

- ANLI (Adversarial Natural Language Inference): ANLI is a challenging NLI benchmark comprising a series of progressively harder rounds in which models must identify the relationship (entailment, contradiction, or neutral) between pairs of sentences.

- WinoGrande: WinoGrande is a large-scale dataset inspired by the original WSC design, but adjusted to improve both the scale and the hardness of the dataset.

- StoryCloze: StoryCloze requires models to read a four-sentence story and then select the correct ending from two options.

- WebNLG: WebNLG corpus consists of sets of RDF triplets describing facts and the corresponding facts in form of natural language text. It challenges models to generate coherent and fluent text paragraphs from triplets.

- XSum (Extreme Summarization): XSum is a dataset designed for evaluating automatic text summarization. It requires models to create a single-sentence summary of a news article.

**Template for XSum: "DOC write summary of above"**

---

Input:
```
The full cost of damage in Newton Stewart, one of the areas worst
affected, is still being assessed.  Repair work is ongoing in
Hawick and many roads in Peeblesshire remain badly affected by
standing water.
(ellipsis)
"Obviously it is heart-breaking for people who have been forced
out of their homes and the impact on businesses." He said it was
important that "immediate steps" were taken to protect the areas
most...

===

Write a summary of the text above :  Clean-up operations are
continuing across the Scottish Borders and Dumfries and Galloway
after flooding caused by Storm Frank.

[FULL ARTICLE OF TEST INPUT]

===

Write a summary of the text above :
```

Output:
```
[MODEL GENERATED SUMMARY]
```

**Template for WebNLG: "Explicit Graph Description"**

---

Input:
```
Take the following graph comprising triple sets, where each
element of a triple is separated by "|" and each triple set by
",":  Aarhus Airport | cityServed | "Aarhus, Denmark".
Make a verbalization of the triple set into plain text.
The Aarhus is the airport of Aarhus, Denmark.

Take the following graph comprising triple sets, where each
element of a triple is separated by "|" and each triple set by
",":
[TRIPLET SET OF TEST INPUT]
Make a verbalization of the triple set into plain text.
```

Output:
```
[MODEL GENERATED VERBALIZATION]
```

Figure 4: **Selected PromptSource templates for evaluating the XSum and WebNLG tasks.** Both examples describe the prompt for one-shot learning scenarios.

