# OpenReview forum: "Exploiting the Potential of Seq2Seq Models as Robust Few-Shot Learners"
_ICLR.cc/2024/Conference — Submitted to ICLR 2024_

### Official Review · Reviewer_SaZz · 2023-10-31

**Soundness:** 3 good
**Presentation:** 3 good
**Contribution:** 3 good
**Rating:** 6
**Confidence:** 3

**Summary:**

This paper explores the potential of Seq2Seq models as robust few-shot learners. A few studies have demonstrated the feasibility of few-shot learning with seq2seq models; however, this has been limited to tasks that align well with the seq2seq architecture, such as summarization and translation. The paper proposes two methods to more effectively elicit in-context learning ability in seq2seq models: objective-aligned prompting and a fusion-based approach. Remarkably, the approach outperforms a decoder-only
model that is six times larger and exhibits significant performance improvements compared to conventional seq2seq models across a variety of settings.

**Strengths:**

1. This paper is well-written regarding the language and organization.
2. The experimental evaluation validates their claims.
3.  The paper performs few-shot learning on a variety of tasks, indicating that the seq-to-seq model can have certain advantages, which is indeed a contribution.
4. Their analysis in the experimental parts is comprehensive.

**Weaknesses:**

1. The paper might want to explain a bit more about the specific tasks of few-shot learning.
2. The paper should explain why the seq-to-seq model is powerful in related tasks, from a machine-learning perspective.
3. Similarity, the paper might want to analyze in-depth why early fusion sometimes yields better performance than late fusion, from a machine-learning perspective.

**Questions:**

No other question, but the model proposed seems too simple. However, the experimental analysis and finding is nontrivial.

---

> ### Author Response · Authors · 2023-11-20
> **Response to Reviewer SaZz**
>
> Thank you, Reviewer SaZz, for your positive opinion. In the following, let us try to address your concerns.
>
> > The model proposed seems too simple. However, the experimental analysis and finding is nontrivial.
>
>
> Thanks again for recognizing the value of our findings. Our work's primary contribution, as recognized by the reviewers, is the rediscovery of in-context few-shot learning capabilities within seq2seq models, realized through comprehensive experimentation. Our endeavor aims to highlight that seq2seq models inherently possess the capability for in-context learning, rather than introducing a novel methodology. Therefore, instead of employing specialized and potentially non-generalizable methods, we focus on applying straightforward and universally applicable methods that are tailored to the innate structure and objectives of seq2seq models. The effectiveness of our proposed objective-aligned prompting and fusion-based approach, while seemingly straightforward, is attributed to their simple yet carefully crafted design, allowing for seamless integration within the seq2seq framework.
>
> Additionally, a critical aspect of in-context learning capabilities lies in the model itself, which needs to be trained with high-quality data and a well-defined objective to possess higher-order skills. Recently, research focus has shifted predominantly towards decoder-only models, resulting in a relative neglect of seq2seq models in the area of in-context learning. Recognizing this gap, we employed a direct yet impactful strategy to showcase the capabilities of seq2seq models, which we consider to be a meaningful endeavor. Through our work, we expect that seq2seq models will be more deeply researched in the future as Large Language Models. We would be grateful if this aspect is acknowledged as our contribution.
>
> ---
>
> > The paper might want to explain a bit more about the specific tasks of few-shot learning.
>
> Thank you for pointing it out. We acknowledge that Section 4 of the main paper lacks detailed explanations of each task. To address this, we will include these details in the Appendix of the revised paper.
>
> ---
>
> > The paper should explain why the seq-to-seq model is powerful in related tasks, from a machine-learning perspective.
>
> In the comparison between encoder-decoder and decoder-only models, a key inductive bias of the encoder-decoder models is their bidirectional encoding capability, especially evident in the processing of examples and queries. This bidirectionality facilitates a deeper understanding of the relationships between examples and queries, allowing the encoder to more precisely extract relevant information and the decoder to generate outputs more effectively based on this information. Furthermore, as highlighted in Section 5.4 of the main paper, encoder-decoder models have a strength in handling permutation bias, inherent to their architecture, meaning their performance is less affected by variations in the order of examples.
>
> ---
>
> > Similarly, the paper might want to analyze in-depth why early fusion sometimes yields better performance than late fusion, from a machine-learning perspective.
>
> Generally, the pretraining objective of encoder-decoder models is designed such that the decoder aggregates the outputs of the encoder. This process maximizes the joint probability of the input text sequence for the decoder, achieved through the encoder-decoder attention module. In this context, the early-fusion method implicitly selects examples that assist in resolving the test query by fully leveraging the encoder-decoder attention module. On the other hand, an encoder-decoder model employing a late-fusion method does not differentiate whether individual examples aid in solving the test query; it simply aggregates all responses. In essence, the late-fusion method does not fully utilize the encoder-decoder attention module. We think that this distinction sometimes enables the early-fusion method to outperform the late-fusion method. We will add these details in Section 5 of the revised paper.

---

### Official Review · Reviewer_aYaC · 2023-11-01

**Soundness:** 3 good
**Presentation:** 3 good
**Contribution:** 3 good
**Rating:** 6
**Confidence:** 3

**Summary:**

An extensive evaluation of zero-shot to few-shot performance of seq2seq models across a wide range of evaluation set is presented. The authors make a case for strong seq2seq model performance for generation and understanding tasks when compared to decoder only models.

**Strengths:**

The primary strength of this works seems to come from experimentally demonstrating that the seq2seq model can outperform the decoder-only model with 6 times larger parameters across diverse datasets.

**Weaknesses:**

Would've liked to see some evaluations around more varied generative tasks like Math/Coding which are more practically useful.

**Questions:**

Are there any tasks where the seq2seq few shot performance was inferior to decoder only models?

---

> ### Author Response · Authors · 2023-11-20
> **Response to Reviewer aYaC**
>
> We thank Reviewer aYaC for your positive opinion. Below are our thoughts to further address your questions.
>
> > Would've liked to see some evaluations around more varied generative tasks like Math/Coding which are more practically useful. Also, are there any tasks where the seq2seq few shot performance was inferior to decoder only models?
>
> Thank you for the insightful question. As you suggested, assessing emerging abilities like math or coding could be a valuable indicator of the practical utility of encoder-decoder models, extending beyond conventional language understanding and generative tasks. However, our method depends on pretrained models, and currently, seq2seq pretrained models face two main issues.
>
> Firstly, they haven’t received as much attention as decoder-only models recently, leading to a lack of models trained on knowledge-intensive data. Our baseline, T5, is solely trained on Common Crawl data, which does not include math and code datasets, making it challenging to reveal these capabilities through in-context learning. For instance, GSM8k score of ****our model(1.6)**** is higher than that of **OPT-66B(1.14)** reported on the Open LLM Leaderboard [1]. However, meaningful comparisons are difficult unless knowledge-intensive datasets are included in the pretrained data.
>
> Secondly, the pretraining objectives of existing seq2seq models aren’t ideally configured for free response generation; their focus was more on improving downstream task capabilities through span corruption and local reconstruction objectives, rather than on language modeling objectives. Consequently, for generative tasks, they might be less effective compared to decoder-only models. To illustrate this, we conducted comparisons with decoder-only baselines on representative generation tasks, XSum and WebNLG, with the results detailed below.
>
> **XSum**
> | Model    | 1-shot (R1/R2/RL) |       5-shot (R1/R2/RL)       |
> |----------|:-----------------:|:-----------------------------:|
> | T5*      |  13.72/2.46/11.97 |         7.57/0.66/6.34        |
> | T5       |  25.12/8.69/20.72 |        26.39/8.99/21.59       |
> | T5-early |         -         | **30.31**/**11.82**/**25.24** |
> | BLOOM-7B |  21.50/4.75/16.33  |        21.96/5.06/17.04       |
> | OPT-13B  |  28.37/9.93/22.46 |       31.32/11.52/24.72       |
> | OPT-30B  |  27.61/10.06/22.20 |       31.54/12.06/25.19       |
> | OPT-66B  | 29.31/10.64/23.45 | **32.52**/**12.86**/**26.19** |
>
> **WebNLG**
> | Model    | 1-shot (R1/R2/RL) |       32-shot (R1/R2/RL)      |
> |----------|:-----------------:|:-----------------------------:|
> | T5*      |  11.44/4.85/10.05 |         0.73/0.12/0.72        |
> | T5       |  49.20/29.22/40.79 |        39.33/23.75/32.72      |
> | T5-early |         -         | **51.11**/**31.16**/**42.52** |
> | BLOOM-7B |  62.77/38.54/51.19|        68.07/43.97/56.10       |
> | OPT-13B  |  55.03/33.04/45.53|       66.26/43.44/55.85       |
> | OPT-30B  |  58.00/35.35/47.78|       66.57/43.07/55.22       |
> | OPT-66B  |  61.44/37.54/50.53| **68.77**/**45.37**/**57.75** |
>
> The asterisk(\*) on the right side of the T5 denotes the case where the sentinel tokens are not used during inference time. R1, R2, and RL denote ROUGE-1,2,L, respectively. We highlight the scores of both our model and the one exhibiting the best performance. Our best model, T5-early, outperforms OPT-13B in terms of ROUGE-2 and OPT-30B in terms of ROUGE-L on the XSum dataset, despite having only 11B in size. However, it shows slightly lower overall performance compared to OPT-66B. For the WebNLG dataset, our model generally scores lower than decoder-only models. Notably, the significant difference in 5-shot performance between T5\* and our proposed T5-early model indicates that our method meaningfully enhances the generation performance of the seq2seq model, irrespective of the pretrained language model’s performance. As previously mentioned, with improvements in the seq2seq model’s pretraining data and objectives, we expect a considerable boost in performance. We will include these experiments in the revised paper and mention the limitations.
>
> [1] https://huggingface.co/spaces/HuggingFaceH4/open_llm_leaderboard

---

### Official Review · Reviewer_6VXy · 2023-11-01

**Soundness:** 3 good
**Presentation:** 3 good
**Contribution:** 2 fair
**Rating:** 5
**Confidence:** 3

**Summary:**

This paper pays attention to the in-context few-shot learning capabilities of seq2seq models. Specifically, this paper conducts comprehensive experiments with an in-context evaluation toolkit to investigate the performance of seq2seq models in few-shot scenarios. In addition, an objective-aligned prompting strategy and a fusion-based approach are proposed. Through extensive experiments, some interesting conclusions are also obtained.

**Strengths:**

1.	This paper is well organized and easy to follow.
2.	The motivation is reasonable and experiments are abundant.
3.	The findings and conclusions about in-context few-shot learning capabilities of seq2seq models will be interesting to the community.

**Weaknesses:**

Several main concerns are as follows:

1.	This paper claims that the objective-aligned prompting strategy is its one key contribution. However, this strategy seems to be very straightforward and some recent state-of-the-art works have already introduced such a strategy. In this sense, this contribution is somewhat limited.

2.	The second contribution of this work is a fusion-based approach, which also comes from the existing works, such as RAG and Fid. Therefore, what’s the main difference and contribution of this work? In addition, in the abstract, the sentence “our approach outperforms a decoder-only model that is six times larger…” shows that the proposed models will be much larger than the competitors. Is it not a significant limitation?

3.	Can we consider the proposed fusion-based approach as a simple ensemble strategy? If so, the authors may need to explain more for this part.

4.	Are there any evidences to support the hypothesis in the sentence of “We hypothesize that the encoding of relations between demonstrations does not significantly impact in-context learning performance.”?

**Questions:**

Please kindly refer to the above comments.

---

> ### Author Response · Authors · 2023-11-20
> **Response to Reviewer 6VXy (Part 1/2)**
>
> Thank you for the detailed review and helpful feedback. We are also pleased that you found our paper as well-organized, reasonable, and interesting to the NLP community. In the following, let us try to address your questions.
>
> > The contribution comes from objective-aligned prompting and fusion-based approach is somewhat straightforward and limited. Some recent state-of-the-art works have already introduced strategies similar to object-aligned prompting. Also, what’s the main difference and contribution of this work compared to FiD and RAG?
>
> Our work's primary contribution, as recognized by the reviewers, is the rediscovery of in-context few-shot learning capabilities within seq2seq models, realized through comprehensive experimentation. Our endeavor aims to highlight that seq2seq models inherently possess the capability for in-context learning, rather than introducing a novel methodology. Therefore, instead of employing specialized and potentially non-generalizable methods, we focus on applying straightforward and universally applicable methods that are tailored to the innate structure and objectives of seq2seq models. The effectiveness of our proposed objective-aligned prompting and fusion-based approach, while seemingly straightforward, is attributed to their simple yet carefully crafted design, allowing for seamless integration within the seq2seq framework.
>
> In exploring the objective-aligned prompting strategy, we realized that within the seq2seq architecture, there are various ways for structuring inputs. While the approach of simply listing examples as inputs aligns well with the objectives of decoder-only models, it may not be as intuitive for currently trained seq2seq models. We observed that in existing studies, this aspect has often been overlooked, with details of how examples are input remaining undisclosed. Therefore, through our experiments with various ablations (for the first time in our knowledge), we have concluded that aligning with the model’s pretraining objective can indeed yield the most effective results.
>
> In the exploration of the fusion-based approach, we particularly focused on one of the significant structural advantages of seq2seq models: the capability to configure the encoding part in parallel in an intuitive manner. Consequently, we indeed adopted methods from FiD and RAG. While following the design of this previous work, we diverged in purpose by using multiple encoders instead of retrieval and scoring modules and applied this to the encoder-decoder few-shot scenario. As a result of effectively customizing modules that previously served entirely different functions, we derived novel insights and achieved impressive performance gains. We would be grateful if this aspect is acknowledged as our contribution.
>
> ---
>
> > The sentence “our approach outperforms a decoder-only model that is six times larger…” shows that the proposed models will be much larger than the competitors. Is it not a significant limitation?
>
> As shown in Table 3 of the main paper, our method, utilizing the T5-11B model, surpasses the performance of the OPT-66B model. This means that our approach yields better results than a decoder-only model that is six times larger, indicating that our model is much smaller than the competitors.
>
> ---
>
> > Can we consider the proposed fusion-based approach as a simple ensemble strategy? If so, the authors may need to explain more for this part.
>
> As per your observation, It seems plausible that our approaches could be interpreted as an ensemble strategy. Following your insightful advice, we will add an interpretation from an ensemble perspective in Section 3 of the revised paper. Thank you for your perceptive and valuable advice.

---

> ### Author Response · Authors · 2023-11-20
> **Response to Reviewer 6VXy (Part 2/2)**
>
> > Are there any evidences to support the hypothesis in the sentence of “We hypothesize that the encoding of relations between demonstrations does not significantly impact in-context learning performance.”?
>
> Thank you for raising a great question. The key reason why increasing the number of shots improves performance in in-context few-shot learning is that it allows for better generalization of the pattern necessary to answer a given question, by repeatedly exposing the model to multiple examples similar to the query, rather than learning the relationship between the examples.
>
> To demonstrate this, we conducted experiments to assess performance using an 8-shot learning scheme across four benchmarks: CB, RTE, WSC, and COPA. CB and RTE are designed to evaluate the ability to infer relationships between two sentences, WSC tests commonsense reasoning ability, and COPA measures causal reasoning and sentence completion skills. Our ablation studies included configurations of 1-shot * 8-encoders, 2-shots * 4-encoders, and 4-shots * 2-encoders, aiming to evaluate the impact of the relationship among few-shots within a single encoder. To ensure a fair experiment that neutralizes the effect of the length extrapolation, we designed the experiments so that the average length entering an encoder does not exceed 512 tokens, which is the maximum pretrained length for a T5 encoder. However, we observed maximum input lengths of 620 for the RTE-4 shot and 526, 617, and 763 for the CB-1,2,4 shot, respectively.
>
> | T5-early     |  CB  |  RTE  |  WSC  |  COPA  |
> |--------------|:----:|:-----:|:-----:|:------:|
> | _1-shot*8-enc_ |****76.79****|****84.00****|****64.26****|****67.31****|
> | _2-shot*4-enc_ |****76.79****|82.00|60.29|54.81|
> | _4-shot*2-enc_ |55.36|82.00|55.23|50.00|
>
> | T5-late      |  CB  |  RTE  |  WSC  |  COPA  |
> |--------------|:----:|:-----:|:-----:|:------:|
> | _1-shot*8-enc_ |****75.00****|****83.00****|****64.62****|****62.50****|
> | _2-shot*4-enc_ |71.43|82.00|61.01|48.08|
> | _4-shot*2-enc_ |53.57|80.00|54.51|49.04|
>
> We highlight the scores for each task that exhibits the best accuracy. Our methodologies, separating each shot into different encoders, attained the highest scores in all scenarios. This experiment supports our assertion that the relationship between few-shot examples does not significantly influence the performance.

---

> > ### Comment · Reviewer_6VXy · 2023-11-22
> > **Thanks for the response**
> >
> > Thanks for the authors' detailed response. I think most of my concerns have been addressed. Unfortunately, I am not very familiar to the NLP field, so I will discuss with other reviewers to obtain the final score. Thank you.

---

> ### Author Response · Authors · 2023-11-23
>
> Thank you for your positive feedback! We have rigorously addressed all of your concerns and clarified points of confusion in our detailed rebuttal. We would also be delighted to provide any further clarifications. As the discussion period draws to a close, we kindly request your consideration in revising the score for our work. Thank you.

---

### Official Review · Reviewer_6fMx · 2023-11-04

**Soundness:** 2 fair
**Presentation:** 2 fair
**Contribution:** 2 fair
**Rating:** 5
**Confidence:** 5

**Summary:**

This paper performs a first-ever extensive experiment comparing the in-context few- shot learning capabilities of decoder-only and encoder-decoder (seq2seq) models on a broad range of tasks. The authors further propose two methods to more effectively elicit in-context learning ability in seq2seq models: objective-aligned prompting and a fusion-based approach. They show their methods significantly outperform decoder-only models.

**Strengths:**

+ This work develops an in-context evaluation toolkit for seq2seq models and conduct extensive experiments to investigate the performance of seq2seq models in zero-shot to few-shot scenarios.

+ The author explore prompting strategies and fusion-based approaches in encoder-decoder models, which reveals their ability of zero/few-shot learning.

+ The comprehensive experiments of comparison between decoder-only and encoder-decoder models could be very useful for researchers in this field.

**Weaknesses:**

- The technical novelty of this work is a bit weak. The proposed objective-aligned prompting and fusion-based approach are straightforward.

- The detailed description of the objective-aligned prompting method is missing.

**Questions:**

See weaknesses.

---

> ### Author Response · Authors · 2023-11-20
> **Response to Reviewer 6fMx**
>
> We sincerely appreciate the reviewer’s constructive comments on our work. Below, we provide our responses to further address your concerns.
>
> > The technical novelty of this work is a bit weak. The proposed objective-aligned prompting and fusion-based approach are straightforward.
>
> Our work's primary contribution, as recognized by the reviewers, is the rediscovery of in-context few-shot learning capabilities within seq2seq models, realized through comprehensive experimentation. Our endeavor aims to highlight that seq2seq models inherently possess the capability for in-context learning, rather than introducing a novel methodology. Therefore, instead of employing specialized and potentially non-generalizable methods, we focus on applying straightforward and universally applicable methods that are tailored to the innate structure and objectives of seq2seq models. The effectiveness of our proposed objective-aligned prompting and fusion-based approach, while seemingly straightforward, is attributed to their simple yet carefully crafted design, allowing for seamless integration within the seq2seq framework.
>
> Additionally, a critical aspect of in-context learning capabilities lies in the model itself, which needs to be trained with high-quality data and a well-defined objective to possess higher-order skills. Recently, research focus has shifted predominantly towards decoder-only models, resulting in a relative neglect of seq2seq models in the area of in-context learning. Recognizing this gap, we employed a direct yet impactful strategy to showcase the capabilities of seq2seq models, which we consider to be a meaningful endeavor. Through our work, we expect that seq2seq models will be more deeply researched in the future as Large Language Models. We would be grateful if this aspect is acknowledged as our contribution.
>
> ---
>
> > The detailed description of the objective-aligned prompting method is missing.
>
> Thank you for the valuable suggestion. Objective-aligned prompting is an abstract concept designed to maximize performance during inference by aligning the prompting design with a model’s pretraining objective. Owing to its dependency on pretrained models, we chose to illustrate this concept through practical examples from well-known models (e.g., T5, UL2), rather than providing a theoretical description of the method. However, as you rightly pointed out, the third paragraph of Section 2 lacked a detailed explanation of the connection between a model’s pretraining objectives and objective-aligned prompting. In response, we will enrich the content of Section 2 in the revised paper, offering a more detailed description of this connection.

---

### Author Response · Authors · 2023-11-20
**General Response to All Reviewers**

We would like to thank all reviewers for taking the time to review our work and for providing constructive feedback. It was encouraging to see that the reviewers regarded our paper as well-written (6VXy, SaZz), considered the findings to be nontrivial and beneficial to the community (6fMx, 6VXy, SaZz), and acknowledged that our extensive experiments strengthen the credibility of our claims (6fMx, 6VXy, aYaC, SaZz).

We have addressed the concerns and suggestions of each reviewer in our reviewer-specific responses. The valuable suggestions from the reviewers have substantially enhanced our draft. The modifications made in the updated version are summarized below, and all changes are highlighted in red in the revised paper.

- [Reviewer 6fMx] In Section 2, we enriched the explanation of objective-aligned prompting.
- [Reviewer 6VXy] In Section 3, we added an interpretation from an ensemble perspective.
- [Reviewer aYaC] In the Appendix E, we added experiments comparing encoder-decoder models with decoder-only models in generative tasks and discussed their limitations.
- [Reviewer SaZz] In the Appendix I, we provided a more detailed explanation of each benchmark task.
- [Reviewer SaZz] In Section 5.2, we added an interpretation of the early-fusion method’s superiority over the late-fusion method from an machine-learning perspective.

---

### Meta-Review · Area_Chair_218b · 2023-12-06

**Metareview:**

This paper studies the in-context learning for encoder-decoder seq2seq models on a range of tasks and also proposes two in-context prompting methods for encoder-decoder models and the experiment results show performance improvement over decoder-only models. This paper has received 4 reviews. Although reviewers recognized the exploration of in-context learning in encoder-decoder models, some reviewers also expressed concerns about the limited technical novelty of in-context prompting(Reviewer 6fMx, 6VXy) as the main technical invention of objective-aligned strategy is a straightforward extension of recent works and fusion-based prompting technique is from existing works like RAG and FiD (Reviewer 6VXy). Reviewer SaZz also mentioned the lack of theoretical analysis of the proposed method and the effectiveness of encoder-decoder models in considered tasks. After reading the paper and discussions, AC also shares a similar feeling that the technical innovation is limited and wants to mention that the use of the OPT-66B model, which is not considered one of the strongest decoder-only models, potentially lowers the soundness of the statement ‘with proper prompting, encoder-decoder model performs better than a 6-times larger decoder-only model’. Considering the agreement of limited novelty among part of the reviewers, this paper is still worth further improvement.

**Justification For Why Not Higher Score:**

The paper did not receive a higher score primarily due to concerns about the limited technical novelty. Reviewers noted that the main technical invention, the objective-aligned strategy, is a straightforward extension of recent works, and the fusion-based prompting technique draws from existing works like RAG and FiD. Furthermore, there was a lack of theoretical analysis of the proposed method and questions about the effectiveness of encoder-decoder models in the considered tasks. These factors led to the conclusion that the paper requires further improvement for a higher score.

**Justification For Why Not Lower Score:**

N/A

---

### Decision · Program_Chairs · 2024-01-16

Reject